Comparative evaluation of the sleep quality metrics between a cardboard bed and a camp cot: a randomized controlled crossover study

http://orcid.org/0000-0001-5785-0426 Hamanishi Seiji 1 2 hamanishi@kusw.ac.jp
Miki Airi 2
http://orcid.org/0000-0002-2842-3561 Sasaki Shinsuke 3
1 Graduate School of Nursing/Nursing Faculty, Kansai University of Social Welfare , Ako, Hyogo , Japan
2 Nursing Faculty, Kansai University of Social Welfare , Ako, Hyogo , Japan
3 Graduate School of Health and Welfare Science/Faculty of Health and Welfare Science, Okayama Prefectural University , Soja, Okayama , Japan
Chen Chong
Electronic publication date: 2024 May 24
Publication date: 2024
Volume: 12
Electronic Location ID: e17392
Received 2024 Jan 2; Accepted 2024 Apr 23
Copyright: © 2024 Hamanishi et al.
Copyright year: 2024
Copyright holder: Hamanishi et al.
License: This is an open access article distributed under the terms of the Creative Commons Attribution License, which permits unrestricted use, distribution, reproduction and adaptation in any medium and for any purpose provided that it is properly attributed. For attribution, the original author(s), title, publication source (PeerJ) and either DOI or URL of the article must be cited.
License URL: https://creativecommons.org/licenses/by/4.0/

Keywords: Evacuees, Emergency beds, Evacuation shelter, Sleep quality, Musculoskeletal strain, Body pressure distribution, Cardboard beds, Camp cots, Ergonomics

Funding: Japan Society for the Promotion of Science (JSPS) KAKENHI 20K19264 This work was supported by the Japan Society for the Promotion of Science (JSPS) KAKENHI Grant Number 20K19264. The funders had no role in study design, data collection and analysis, decision to publish, or preparation of the manuscript.

==============================
Background

Health-beneficial emergency bedding has become increasingly important for dealing with natural disasters such as the anticipated Nankai Trough earthquake in Japan. When the Great East Japan Earthquake occurred, cardboard beds were provided to evacuees. However, there were concerns about lower back pain and sleep disturbances, as cardboard beds offer insufficient pressure distribution. This study aimed to compare the effects of cardboard beds with those of foldable camp cots on sleep quality.

Methods

A randomized controlled crossover study involving 20 healthy participants aged 18–45 years was conducted between June 2022 and January 2023. Participants were asked to sleep for one night on a camp cot and for another night on a cardboard bed, with a minimum three-day washout period between the two nights. Body pressure distribution and sleep metrics obtained from polysomnography (PSG) and questionnaires were compared between the two-bed types (P < 0.05).

Results

The camp cot exhibited better body pressure distribution than a cardboard bed, leading to improved sleep satisfaction, bedding comfort, and reduced morning sleepiness. Nevertheless, polysomnography revealed no notable differences in sleep metrics or sleep architecture between the two types of beds.

Conclusions

Our findings indicate that cardboard beds have lower pressure dispersion capabilities than camp cots, leading to an increased number of position changes during sleep. Additionally, subjective sleep quality, such as alertness on waking, sleep comfort, and sleep satisfaction, was lower for cardboard beds, suggesting that camp cots might offer a more comfortable bedding option for evacuees. However, there were no discernible differences between the two-bed types in terms of objective sleep metrics derived from PSG. The potential for sleep disturbances caused by lower back pain from a hard mattress has been noted, and it is possible that a single night’s experience in healthy individuals might not be enough for sleep issues to manifest.

Introduction

In recent years, large-scale natural disasters have become a global concern and have significantly impacted people’s lives. This problem is particularly acute in Japan, where a Nankai Trough earthquake would probably result in approximately 5 million evacuees (Cabinet Office, 2013). One problem during disasters is the difficulty of providing appropriate bedding to evacuees. According to the guidelines developed by the Cabinet Office, Japanese municipalities are not required to stockpile bedding other than blankets, which means that evacuees often sleep directly on the floors of classrooms and school gymnasiums. To address the bedding problem, providing emergency cardboard beds for evacuees only began during the Great East Japan Earthquake (Nara et al., 2013). Although cardboard beds provide some insulation from the cold floor, our previous research suggested problems regarding pressure dispersion on cardboard beds (Mizuno et al., 2017; Hamanishi, 2021; Hamanishi et al., 2023). As too-firm mattresses induce lower back pain and sleep disruptions, body pressure concentration due to inappropriate bedding increases the likelihood of sleep disorders in evacuees (Jacobson et al., 2010).

In a large-scale disaster, infrastructure and transportation networks can be widely destroyed, thereby limiting the supply of materials immediately after the disaster. During the Kumamoto earthquake, 60% of the municipalities in need did not receive a supply of cardboard beds (Sueta et al., 2019). Since the pressure distribution capacity of the cardboard bed itself is not high, it must be used with a futon or an appropriate mattress (Hamanishi, 2021; Hamanishi et al., 2023). While agreements exist to provide cardboard beds to Japanese municipalities, there are no agreements to provide mattresses and futons, making them potentially more challenging to supply than cardboard beds (Mizutani, 2012; Japan Corrugated Case Association (JCCA), 2023). In addition to cardboard beds, folding camp cots are being used as emergency beds. Owing to their lightweight nature, camp cots can be stored compactly, rendering them suitable for individual stockpiling. Moreover, the fabric surface of the camp cot is more pliable than that of the corrugated cardboard bed, suggesting that the pressure distribution capacity of the camp cot may exceed that of the cardboard bed. However, the pressure dispersion capacity and sleep improvement effects of cots and cardboard beds have not been adequately investigated.

Therefore, this study aims to compare between camp cots and cardboard beds for disaster use regarding pressure distribution and sleep indicators. Clarifying these aspects could provide scientific evidence to recommend adequate emergency beds in preparation for large-scale disasters.

Materials and Methods

Study participants

In this study, the sample size was determined based on an effect size = 0.7, α = 0.05, and a power (1-β) = 0.80, using G*Power (Heinrich-Heine-Universität Düsseldorf, Düsseldorf, Germany), resulting in a total of 20 participants from June 2022 to January 2023. Healthy participants were recruited through our website. The two-day experimental schedule was determined in accordance with the individual schedules of the research participants. Particularly, to accommodate the sleep pattern variations associated with a menstrual cycle, the study dates for female participants were scheduled within two weeks following the end of their last menstrual period. If it was not feasible for a participant to join the experiment during the same menstrual cycle, an alternative date was set in the subsequent menstrual cycle. As shown in the flowchart in Fig. 1. The inclusion criteria were as follows: (1) age ≥18 and ≤50 years, (2) ability to provide informed consent, and (3) absence of medical interventions for sleep disorders. The exclusion criteria were as follows: (1) presence of irregular sleep-wake rhythm, (2) taking medicines for psychiatric or sleep disorders, (3) night worker or shift worker, (4) visiting areas with a time difference of 3 h or more, (5) pregnancy or breastfeeding, (6) people who have a disease that may change suddenly, (7) physical symptoms that interfere with sleep, (8) sleep disorders or chronic daytime sleepiness, and (9) current smokers. The planned recruitment target of 20 participants has been achieved, leading to the conclusion of participant enrollment for the study.

Figure 1 Consort flow diagram.

Design and protocol

The study protocol is illustrated in Fig. 2. This was a randomized controlled crossover study, and data were collected from June 2022 to January 2023. All participants provided informed consent to participate in the study and were assigned to sequences A (n = 10) or B (n = 10) using random numbers calculated in Microsoft Excel (Microsoft, Redmond, WA, USA). The order of bed types for the participants allocated to Sequence A was as follows: 1st, a camping cot; 2nd, a cardboard bed. The order for participants allocated to Sequence B was 1st, a cardboard bed; 2nd, a camping cot. A washout period of at least three days was implemented between the two experimental conditions. Participants were requested to refrain from consuming alcohol for the entire day, caffeinated beverages after 16:00, and any food after 21:00 on the day of the experiment. Participants were required to sleep in a room arranged by the research team to maintain controlled experimental conditions. Compliance with these conditions was verified by the researchers, who also confirmed that participants had completed a pre-sleep survey. Researchers provided explanations on the methodology of electrode application and device handling. Additionally, participants were given QR codes linking to instructional videos, ensuring they could independently operate the devices. After awaking, research staff verified that participants had correctly understood and adhered to the experimental procedures, device operation, and any precautions during the experiment. They also confirmed that participants had responded to all post-awakening questions.

Figure 2 Experimental protocol.

Sleeping conditions

On the day of the experiment, researchers arranged the room’s environment and prepared the bed. As shown in Fig. 3, over the course of two experimental days, the participants were asked to sleep on (A) a foldable camp cot (Brooklyn Outdoor Company, Brooklyn, NY, USA) on one day and on (B) an emergency cardboard bed (BRAIN, Osaka, Japan) on the other. On both days, a single blanket was provided; however, the use of pillows was prohibited. To maintain the sleep conditions for both nights, all subjects were asked to go to bed while wearing the same clothes and at the same time (PM10:00–11:00), and to regulate their room temperature (24–26 °C), humidity (40–60%), and illumination level (0–5.0 lx) during sleep. In addition, the participants were asked to refrain from napping and drinking any caffeinated or alcoholic beverages.

Figure 3 Illustration of the experimental conditions.

Polysomnographic parameters

Sleep parameters on each night were recorded using a portable polysomnography device (Insomnograf; S’UIMIN Inc., Tokyo, Japan). The portable device is reported to have an 86.9% concordance rate and 0.80 kappa coefficient with a typical polysomnography (PSG) device (Horie et al., 2022; Seol et al., 2022). The recording system of this device consisted of five electroencephalogram derivations: Fp1–M2, Fp2–M1, Fp1–average M, Fp2–average M, and Fp1–Fp2. The records were scored every 30 s to classify sleep stages as wakefulness (stage W), non-REM (N1, N2, and N3), or REM (Fig. 4). All participants were instructed to attach electrodes to their heads and push the recording button immediately before going to bed and after waking up.

Figure 4 Portable polysomnographic device.

(A) Insomnograf: This recording device is lightweight (162 g) and easily attached and detached because of the soft-sticking integrated electrodes, which is disposable. (B) The EEG electrodes (Fp1, Fp2) and one reference electrode (Fpz) were attached to the forehead. (C) The EEG electrodes (M1, M2) were attached to the mastoids.

Body pressure distributions

SR Soft Vision (Sumitomo Science & Engineering, Aichi, Japan) was used as the body pressure measurement system (BPMS). The device was equipped with 1,600 (64 × 25) pressure sensors. The size of each sensor was 784 mm2 (28 mm × 28 mm), and the total sensing area of the device was 1,800 mm × 700 mm. The body-mattress contact pressure and body contour area of the whole body in the supine and lateral positions were captured (Fig. 3). Each subject wore identical clothes during all experiments to minimize variations due to clothing. The detection range of the contact pressure was set between 10 and 110 mmHg to preclude the detection of pressure from clothing. The BPMS device recorded pressures at a maximum of 110 mmHg; thus, actual pressures exceeding this threshold were noted as 110 mmHg. Participants were asked to assume both supine and lateral positions on each bed, with the body pressure measured in sequence from a supine position to a lateral position. To calculate the average values, the participants were instructed to lie down in the same position two consecutive times. To ensure accurate measurements, the contact pressure distribution of the model was obtained after a stabilization period of approximately 10 s (Hamanishi, 2021; Hamanishi et al., 2023). After assessing the pressure sensing area 10 times using a dummy model prior to the main experiment, we calculated the measurement error (max−min/mean × 100%), which was 2.7%. The accuracy of this device was verified by the manufacturer in January 2022.

Questionnaire

Prior to the start of the experiment, information on personal attributes and eligibility criteria, such as age, sex, and body mass index (BMI) were collected. Before the start of each experimental day, we checked the participants’ recent sleep status, ensured that their health was suitable for participation, and confirmed whether they had consumed beverages containing alcohol or caffeine. After waking, subjective bed comfort, bed surface firmness, sleep satisfaction, and neck and lower back stiffness were assessed using a visual analog scale (VAS). The VAS range for subjective bed comfort was set at 0 to 10 (most to least comfortable), bed firmness at 0 to 10 (softest to firmest), sleep satisfaction at 0 to 10 (very good sleep to no sleep at all), and lower back stiffness at 0 to 10 (no pain or stiffness at all to worst possible pain or stiffness). Sleepiness upon awakening was evaluated using the Japanese version of the Karolinska Sleepiness Scale (KSS-J).

Data analysis

We used a linear mixed-effects model to compare sleep metrics measured by PSG and the results of the self-administered questionnaire between a camp cot and a cardboard bed. The period and bed type were entered into the model as fixed effects, and subjects nested within the sequence were entered as random effects. This study aimed to investigate the impact of the differences in body pressure distribution capabilities between two types of beds on sleep quality. Therefore, age, gender, and Body Mass Index (BMI), which could act as potential confounding factors affecting body pressure distribution, were incorporated as covariates in the model. Age, sex, and BMI were entered into the model as covariates to control for potential confounding factors. A two-sided test was used for the comparison of the statistical characteristics of the study participants. On the other hand, a one-sided paired T-test was chosen for the comparison of body pressure distribution based on the results of preliminary experiments. In our study, the effect size for t-tests was measured using Cohen’s d, and for the linear mixed models, we adopted R^2 as the measure of effect size (Nakagawa & Schielzeth, 2013). Statistical significance was set at p < 0.05. All data were compiled using Microsoft Excel (Office 365; Microsoft, Bellingham, WA, USA) and statistically analyzed using IBM SPSS (version 28.0; SPSS, Armonk, NY, USA).

Ethics

All participants provided written informed consent before participation. This study was conducted with the approval of the Ethical Review Board of Kansai University of Social Welfare (No. 2-0215). This study was registered with the University Hospital Medical Information Network Clinical Trials Registry (No. UMIN000045165).

Results

Participant characteristics

A total of 23 individuals expressed interest in participating in the study. One individual withdrew from participation and two lost contact with us, resulting in 20 participants who took part in the experiments. No participants dropped out after the experiment commenced, and data from all 20 participants were analyzed (Fig. 1). As shown in Table 1, of the 20 participants enrolled in the study, three were males and 17 were females. The mean age of the participants was 26.75 years with a standard deviation (SD) of 9.46 years, and no significant difference was observed between the groups (P = 0.77). The average body mass index (BMI) was reported as 21.07 kg/m2, with a SD of 2.69, and no difference was found between the groups (P = 0.57). Furthermore, no significant difference was observed in the average sleep duration over the 3 days prior to the experiment (Camp cot; P = 0.08, Cardboard bed: P = 0.18).

Table 1 Characteristics of participants.

	Sequence A (N = 10)	Sequence B (N = 10)	P-value	
Sex (Male/Female)	2/8	1/9		
Age (years old)	26.10 ± 10.46	27.40 ± 3.37	0.77	
BMI (kg/m2)	20.70 ± 2.00	21.42 ± 3.37	0.57	
Average sleep time over the three nights before the experiment (h)		
Camp cot	7.03 ± 0.22	6.91 ± 0.23	0.08	
Cardboard bed	6.94 ± 0.31	7.05 ± 0.13	0.18	
Daytime sleepiness before the experiment		
Camp cot	0/10	0/10		
Cardboard bed	0/10	0/10		
Note:

Age, BMI, and Average sleep time (over the 3 days before the experiment) is presented as Mean ± SD.

Difference in body pressure distribution between camp cots and cardboard beds

Figure 5 presents the results of a linear mixed model for repeated ANOVA to compare the body-mattress contact pressure distribution. The results relative to mean contact pressure and contour area showed significant differences between a camp cot and a cardboard bed. The mean body pressure in a supine position was found to be 34% lower on the camping cot compared to the cardboard bed (P = 0.002), and it was also 11% lower in a lateral position (P = 0.045). Comparative analysis also revealed that the body pressure area while lying on a camping cot was 31% larger than that when lying supine on a cardboard bed (P < 0.001), and was 12% larger in a lateral position (P < 0.001).

Figure 5 Comparison of body pressure distribution between camp cots and cardboard beds.

(A) Illustration of body pressure distribution in a spine position. (B) Average body pressure in a supine position. (C) Contour area in a supine position. (D) Illustration of body pressure distribution in a lateral position. (E) Average body pressure in a lateral position. (F) Contour area in a lateral position. Each dot represents an individual data measurement, and lines connecting these dots indicate repeated measurements from the same participant. All data were compared using a linear mixed method (P < 0.05).

Comparison of subjective measures between camp cots and cardboard beds

Figure 6 presents the comparison between camp cots and cardboard beds with regard to self-administered sleep parameters including bedding firmness, cervical and lumbar stiffness, sleep comfort, sleep satisfaction, and daytime sleepiness (KSS). All subjective parameters demonstrated statistically significant differences between the camping cot and the cardboard bed. According to these results, participants reported that the camp cot was perceived as softer (P < 0.001) and more comfortable (P < 0.001) than the cardboard bed, with significantly higher scores in sleep satisfaction (P < 0.001). Furthermore, data indicated that the camp cot induced less cervical (P < 0.001) and lumbar strain (P < 0.001) compared to the cardboard bed, and the sleepiness score (KSS) after waking was lower (P < 0.001).

Figure 6 Comparison of subject sleep quality between camp cots and cardboard beds.

(A) Bedding firmness. (B) Sleep comfort. (C) Cervical stiffness. (D) Lumbar stiffness. (E) Sleep satisfaction. (F) Sleepiness after awaking (KSS). Each dot represents an individual data measurement, and lines connecting these dots indicate repeated measurements from the same participant. All data were compared using a linear mixed model (P < 0.05).

Comparison of sleep-related metrics using portable polysomnography in camp cots versus cardboard beds

Figure 7 presents the outcomes of our investigation into the effects of two different types of beds on various sleep parameters, including time in bed (TIB), total sleep time (TST), sleep efficiency (SE), sleep latency (SL), wake after sleep onset (WASO), and the proportions of sleep stages N1, N2, N3, and REM. Our analysis indicated no significant differences in TIB, TST, and calculated SE between the two bed types. Similarly, the comparison of SL and WASO values across the bed types did not reveal any significant variances. Additionally, there were no significant differences observed in the distribution of sleep stages, including N1, N2, N3, and REM stages, among the participants who slept on the two types of beds. However, the number of tosses and turns per hour was significantly lower in participants using the camp cot (P = 0.03).

Figure 7 Comparison of sleep metrics using PSG between camp cots and cardboard beds.

(A) Time in bed (TIB). (B) Total sleep time (TST). (C) Waking after sleep onset (WASO). (D) Sleep efficiency (SE). (E) Sleep latency (SL). (F) Frequency of turning over. (G) Proportion of sleep stage. Each dot represents an individual data measurement, and lines connecting these dots indicate repeated measurements from the same participant. All data were compared using a Linear Mixed Model (P < 0.05).

Discussion

In this study, we investigated whether camp cots or cardboard beds were better for disaster use by evaluating pressure distribution and sleep indicators with each bed type. We found that camp cots provided significantly higher-pressure dispersion than cardboard beds. Subjective sleep parameters, such as sleep satisfaction, bedding comfort, and sleepiness after awakening, showed more favorable outcomes with the foldable camp cot. Additionally, there were fewer changes in sleep position with the camp cot than the cardboard bed. However, no significant differences were observed in the sleep metrics derived from PSG between the two bed types.

The subjective bedding firmness and comfort of a camp cot were also superior to those of a cardboard bed. Previous research has reported that the body pressure dispersion of emergency cardboard beds is insufficient, and the body pressure distribution when lying on these beds is comparable to that when lying on the floor directly (Hamanishi, 2021; Hamanishi et al., 2023). Using a medium-firm mattress is known to be more suitable than an extra-firm mattress for improving low back pain and sleep quality (Jacobson et al., 2010; Radwan et al., 2015). Our results also suggest that sleeping on a cardboard bed with floor-like firmness for just one night increases cervical and lumbar strain. In this study, the frequency of turning over was significantly lower when sleeping on a camp cot bed than when sleeping on a cardboard bed, suggesting that bed-specific body pressure dispersion affects turning frequency. Turning over during sleep serves various effects, such as redistributing concentrated body pressure and dissipating trapped heat between the mattress and the body. In addition, sleep satisfaction was higher and sleepiness score after awaking were lower when using a camp cot than when using a cardboard bed. From our results, we presumed that the difference in body pressure distribution during sleep plays an important role in the superior subjective sleep indicators of camp cots compared with cardboard beds. Although we expected that differences in objective sleep metrics would occur due to variances in body pressure dispersion, no significant differences were observed between the two types of beds in PSG-evaluated sleep metrics, including TIB, TST, SE, SL, WASO, and the proportion of each sleep stage. One potential explanation for these results lies in the demographic characteristics of the participants. Many participants were young women, with an average age of 26.75 years and a mean BMI of 21.07. Given that body weight and age are risk factors for back pain, the lower weight of participants likely resulted in less cervical and lumbar strain during a single night’s sleep. Musculoskeletal pain, such as back pain, is known to disrupt sleep, but we speculate that the strain on the musculoskeletal system was not significant enough to impact sleep in this young, healthy cohort. Lower back pain has been reported as the most prevalent health issue among evacuees (Ichiseki, 2013; Tsuboi et al., 2022). Furthermore, research has shown that individuals with lower back pain experience a decrease in sleep quality when sleeping on an extra firm mattress (Radwan et al., 2015; Caggiari et al., 2021; Jacobson et al., 2010). The prevalence of lower back pain among evacuees increases over time, but lower back pain was included as an exclusion criterion in our study, resulting in no participants suffering from this condition. Consequently, we assumed that spending just one night on a cardboard bed was not sufficient to induce musculoskeletal strain significant enough to cause sleep problems in young, healthy participants. However, when a major disaster occurs, evacuees often need to use emergency beds for an extended period. This prolonged use of cardboard beds without any mattress could lead to back pain and decrease sleep quality. To explore this further, a longer-term intervention study is essential.

While the findings of this study offer valuable insights, some limitations need to be addressed. First, our study was conducted during a period when the Japanese government called for people to refrain from traveling due to COVID-19. As a result, recruiting study participants with a balanced gender ratio proved challenging, resulting in a predominance of young female participants. To minimize the effects of this biased sample, we utilized a crossover design; however, careful consideration is required in the interpretation of our findings. Additionally, high-risk groups for back pain, such as the elderly and the extremely severe obese, were not included in our study. Different outcomes and insights can be obtained by focusing on these populations. Further studies in different populations are essential to gain a more comprehensive understanding of this issue. Secondly, actual evacuation shelters are influenced by various factors, including fluctuating temperatures, cold temperatures on the floor, ambient noise, and the density of evacuees, all of which can potentially disrupt sleep. However, our study was aimed at elucidating the impact of different mattress firmness on sleep, thus it was conducted in an environment that differs from actual shelter environments. These conditions may not reflect the challenges faced by individuals in actual evacuation scenarios. Furthermore, the participants were asked to follow the prescribed procedures and allowed to eat and take baths. Thirdly, since blinding the types of beds was difficult, both participants and researchers were aware of the differences in them. Therefore, the possibility of bias in the results of this study cannot be denied. The most important limitation of this study was that the intervention lasted only one night. We cannot dismiss the possibility that the first-night effect (FNE), which typically occurs when sleeping in a new place for the first time, may have influenced the sleep data. Although study participants were required to sleep and wake up within a specified time range, some participants woke up earlier than planned and found it difficult to fall back asleep. Since sleep patterns fluctuate daily, it was difficult to determine whether the reduction in sleep duration was due to differences in beds or the unfamiliar environment, but observing sleep over several consecutive nights could potentially clarify the cause. However, since our research was only able to collect data for one night per condition, careful interpretation of the study outcomes is necessary. In the future, we aim to conduct empirical research in settings that simulate actual evacuation shelters more closely, evaluating data from multiple nights of sleep and including participants from a broader age range.

Conclusion

Our findings indicated that using a folding camp cot provides better body pressure distribution and enhances comfort during sleep compared with a cardboard bed. However, the difference in bedding firmness from just one night’s use may not result in measurable changes in sleep architecture.

Supplemental Information

Supplemental Information 1 T test.

Supplemental Information 2 A linear mixed model.

Supplemental Information 3 Questionnaire.

Supplemental Information 4 Dataset.

These data were utilized to compare body pressure distribution and sleep-related metrics when lying on two types of emergency beds.

We would like to thank Editage for English language editing.

Additional Information and Declarations

Competing Interests

Author Contributions

Human Ethics

Data Availability

Clinical Trial Registration

The authors declare that they have no competing interests.

Seiji Hamanishi conceived and designed the experiments, performed the experiments, analyzed the data, prepared figures and/or tables, authored or reviewed drafts of the article, and approved the final draft.

Airi Miki performed the experiments, authored or reviewed drafts of the article, and approved the final draft.

Shinsuke Sasaki performed the experiments, authored or reviewed drafts of the article, and approved the final draft.

The following information was supplied relating to ethical approvals (i.e., approving body and any reference numbers):

The Ethical review board of Kansai University of Social Welfare.

The following information was supplied regarding data availability:

The raw measurements are available in the Supplemental File.

The following information was supplied regarding UMIN-CTR clinical trial registration: UMIN000045165.

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
