# Peer review of "Comparative evaluation of the sleep quality metrics between a cardboard bed and a camp cot: a randomized controlled crossover study"

_PeerJ, doi:10.7717/peerj.17392_

## Round 0.1 · original submission · Major Revisions

As you can see, the reviewers have offered constructive feedback that I believe will greatly assist you in revising your manuscript. I kindly request that you provide comprehensive responses to each comment from the reviewers.

Reviewer 1 ·

Basic reporting

The authors have with their current study contributed to a highly relevant field of research in areas where (natural) disasters lead to massive needs of emergency bedding and specific versions of such beds potentially can harm sleep quality and quantity. In this well structured article, which is written in clear and correct English, the authors compare two such beds in relation to both subjective and objective sleep parameters. The Figures, Tables and raw data that is accompanying the manuscript are clear and appropriate.

Experimental design

The experimental design raises some questions and could be improved taking into account the following:
1. A justification for the age range (18-50 years). Since (natural) disasters by definition hit the population as a whole also younger and older age groups than the selected will have to be investigated. Although difficult to add to the current study at this stage, this should be commented on in the articles's discussion section.
2. The design of having to sleep on each of the beds for only 1 night is a true weakness that considerably limits the value of the results obtained. Sleep has a strong day-to-day variation and more nights on each of the beds would have improved the study a lot. Again, this is a point that is difficult to change at this stage, but should be mentioned in the discussion section more extensively than what is currently done in line 216/217.
3. The authors do mention sleeping on the floor as what would obviously be the worst scenario. It would be interesting, however, to include sleeping on the floor as a third experimental condition, together with the cardboard bed and the camp cot. This could moreover help the convince the responsible authorities to ensure sufficient camp cots, rather than cardboard bed or just floors to sleep on.
4. Sleeping under optimal external conditions at home (as in the current study) is hardly comparable to sleeping in big halls after the occurrence of a (natural) disaster. To increase ecological validity, the authors may wish to discuss the need of future studies to be carried out in big halls as well, where sound, stress and various other parameters may have such a big effect on sleep that the effect of the bed diminishes significantly.
5. Compliance to experimental procedures is always problematic in field studies without experimental supervision. Please mention how it as checked to what extent the instructions as described in lines 79-83 were followed.
6. Some of the questions have methodological issues. Question 1 "Do you have an irregular sleep-wake rhythm" is usually interpreted differently by different individuals. Hence, some follow up question as to quantify this variation would have been needed. Also question 2 that asks for medication affecting mental state or sleep is difficult to answer and also subject to individual variation.
7. An explanation as to why the authors did not aim for an equal number of male and female participants would be good to add.

Validity of the findings

As stated above, although the conclusions are valid based on the results obtained, there are some substantial issues in this earlier stage of result collection that puts a severe limitation on the conclusions. These are:
1) having just one night on each of the beds ignores the naturally occurring daily variation in sleep;
2) the ecological setting of sleeping on these beds at home ignores the potentially much more severe sleep disturbing parameters as sound and stress in big evacuation halls.

Reviewer 2 ·

Basic reporting

I could not find the funding statement.

Experimental design

The experimental design of the present study have following problems.

1) Most of the subjects were females. However, the authors seemed to make no control of the menstral cycle for the data collection.
2) It is unclear if the noctrunal sleep experiments conducted in the laboratory or the volunteers' home. If the experiments were conducted in the volunteers' home, how the authors confirmed their environmental conditions (ambient temperature, ambient humidity and light intensity) during sleep. In case of data collection at home, various artifacts such as noise and/or awakening stimulus made by the other family members could affect the volunteers' sleep. Besides, in case of the experiment conducted in winter season, keeping room temperature between 24 to 26 degrees Celcius seems to be very difficult and is much different from that during daytime.
3) According to Figure 7 and raw data shown in the dataset table, TIB was not controlled. Five volunteers (A,C, D, H, S) showed the diffeernce between the condition of camp cot and cardboard bed more than one hour. Although the time to go to bed and rising in the morning is not presented in the results, sleep phase (timing) might not be well controlled.
4) As for the measurements of body pressure distributions, the method should be described in detail. The order of the body position (supine and lateral) and the duration that the volunteer kept each position for the measuremnts should be described.
5) As for the statistical analysis, the authors used age, sex, and BMI as covariates. For evaluating sleep parameters measured by polysomnography, it is not necessary to use such covariates. Absolute values are more meaningful for evaluation.

Validity of the findings

The results of subjective evaluations shown in Figure 6 is feasible based on the material of each bed. However, as mentioned above, the condition of data collection was not controlled to derive the meaningfull results.

Additional comments

There are following points to be revised or clarified:
1) In the results section (line: 171-174), there is no description to explain the results.
2) In the results section, p values must be shown to explain the significant differences.
3) In figure 6, all of the p values are "P<0.001". It seems that some of the p is above 0.001, for example the panel (E).
4) In figure 7, the note describes "(B) WASO. (C) TST". It looks (B) is TST and (C) is WASO.
5) In line 148, the final description is insuffcient ("According to").
6) In line 146, the volunteers were 3 males and 17 females. However, in the dataset table, the description for SEX shows 19M and 1F.

Reviewer 3 ·

Basic reporting

no comment

Experimental design

- After waking, neck and lower back stiffness and sleepiness upon awakening were measured. Could you provide more details on how these two scales were measured? For example, how many researchers conducted the measures? Were those measures independent? Were the researchers blinded to the allocation group of the participants? If not, please include these in the discussion as the potential limitatins.
- In statistical analysis line 130, please specify whether the p-value was two-sided or one-sided. Could you describe what descriptive statistics were used to summarize continuous and categorical variables?
- Could you compare the demographic characteristics of the participants who were assigned to group A vs. those who were assigned to group B in table 1? If there were no significant differences, this could help confirm that randomization was conducted properly.

Validity of the findings

no comment

---

## Round 0.2 · accepted · Accept

Thanks to the authors for addressing the reviewers' concerns.

Reviewer 1 ·

Basic reporting

Many thanks to the authors for this major revision. All the revised sections look good to me and I have no further comments.

Experimental design

No comment and happy with the revisions being made by the authors.

Validity of the findings

No comment and happy with the revisions being made by the authors.

Additional comments

No comment and happy with the revisions being made by the authors.

Reviewer 3 ·

Basic reporting

The authors have sufficiently addressed my comments.

Experimental design

The authors have sufficiently addressed my comments.

Validity of the findings

The authors have sufficiently addressed my comments.